# Simulation of Biogas Conversion Using Porous Solid Oxide Electrochemical Cells: Virtual Prototyping

**Konakpo Parfait Kamara** [1,2]**, Gérard Merlin** [3]**, Gaoussou Bamba** [2]**, Florence Druart** [1] **and Jonathan Deseure** [1,*]

1    Institute of Engineering and Management, CNRS, Grenoble INP, LEPMI, Université Grenoble Alpes, Université Savoie Mont Blanc, 38000 Grenoble, France
2    UFR SSMT, Université Félix Houphouët-Boigny GPE-Abidjan, Abidjan 01 BP V34, Côte d'Ivoire
3    CNRS, LOCIE, UMR 5271, Polytech Annecy, Université Savoie Mont-Blanc, Chambéry, 73370 Le Bourget du Lac, France
*    Correspondence: jonathan.deseure@lepmi.grenoble-inp.fr

**Abstract:** The computer-aided engineering approach has made it possible to achieve virtual prototypes and to describe expected performances of new apparatuses. In this study, a direct production of syngas with biogas using the configuration of the cascade conversion cell in the supply feed direction of the system was exhibited. Momentum, heat, mass and charge balances were solved using COMSOL Multiphysics® commercial software. These simulations allowed calculation of distributions of partial pressures for all gas species within the anode ($CH_4$, $H_2$, CO, $CO_2$, $H_2O$, $N_2$), as well as velocity field and temperature. The conversion process included methane reforming (steam and dry) associated with the water–gas shift reaction. The computing results showed that the configuration of three porous oxide solid cells based on a solid oxide fuel cell (SOFC) system conferred a larger active surface area and limited thermal stress in oxide materials. In addition, depending on the production process of the biogas, feeding composition strongly influences the conversion rate of $CO_2$ and $CH_4$. We observed that production of syngas was optimal for a $CO_2/CH_4$ ratio = 1.

**Keywords:** biogas conversion; dry methane reforming; steam methane reforming; syngas production; solid oxide electrochemical cell; multiphysics modeling

## 1. Introduction

Nowadays, hydrogen rich-fuel is adopted by most SOFC (solid oxide fuel cell) developers. The SOFC systems allow areal fuel flexibility, due to high operating temperature. These fuel cells can be fed by fossil fuels, natural gas, methanol and ethanol; thus, biogas that contains methane is a fuel of interest for low-carbon-power generation. SOFCs are an efficient technology for promotion of energy security and better energy conversion [1]. Gandiglio et al. [2] studied a plant consisting of three SOFC modules (174 kWe electric nominal capacity) working at 90–100% of full power (electric and heat energy), using a variable biogas production profile. This work has shown effectiveness of the SOFC system in converting biogas, which is comprised of two major compounds (i.e., $CH_4$ and $CO_2$). Biogas production through anaerobic digestion (AD) is the simplest way to extract energy from biomass such as organic waste materials [3]. Methane content of biogas typically ranges from 45% to 75% by volume, with most of the remainder being $CO_2$. In 2018, Europe, the People's Republic of China and the United States accounted for 90% of global biogas production, with a total world production close to 34 Mtoe (395 TWh) [4]. Biogas production is an eco-friendly strategy for power generation from biomass, with a mixed production of methane and hydrogen and a final residue that can be used as a nutrient-rich fertilizer [5]. Moreover, biogas can be also used as a source for production of syngas ($H_2$ and CO: $CH_4$ dry reforming). Jung et al. [6] described the potential future directions of this use of biogas and suggested new paths toward hybrid biological/chemical processes

for syngas production. According to Ren et al. [7], biomass is a promising alternative to syngas production.

Direct internal steam reforming (DIR) or gradual internal reforming (GIR) within the SOFC anode is possible [8]; this methane conversion allows hydrogen production without use of a separate reformer. These reforming reactions, i.e., using water as a gas converter and using $CO_2$ for dry reforming, have similar thermodynamic characteristics. Therefore, due to the large amount of carbon dioxide in biogas, the dry-reforming reaction is a suitable process to convert biogas into hydrogen or synthesis gas (CO and $H_2$). Two way are currently proposed to fully exploit biogas energy systems. According to X. Chen et al. [9], direct heat utilization of $CH_4$ from biogas for power generation and cogeneration (e.g., as an urban gas source) is the most common use. Another method of biogas utilization is to produce another energy carrier, such as $H_2$ for fuel cells or syngas ($H_2$ and CO) for synthetic fuels [9]. Therefore, models and computational methods could be used as cost-effective tools to optimize energy performance of these systems. Many relevant works have been carried out using computational fluid dynamics (CFD) software in three dimensions because multidimensional modeling improves description of gas composition, current density and temperature along gas channels [10]. These studies considered SOFC single-cell or SOFC stacks and highlighted complex interactions between electrochemistry, heat and mass transfer. In addition, some computing works have focused on mass transfer of active chemical species, including electrochemical processes and distribution of electric potential within electrodes and electrolytes, as performed by J. Aicart et al. [11] for co-electrolysis systems. Consequently, an approach that mixes biogas conversion and simulations should be relevant, according to D. Papurello et al. [12]; their study was based on simulations of a SOFC tubular cell system that was fed directly by biomass. These authors have highlighted the significance of conversion of biogas to syngas.

The present work exhibits a system of biogas-to-syngas production using a cascade of three electrochemical half-cells of porous solid oxide. Based on the experimental works of S. George et al. [13], in which the authors observed the electrothermal boost of biogas conversion into syngas, this study involved a computer-aided engineering approach [14] to improve syngas production. Computational fluid dynamics can produce numerical results in order to improve chemical performances of apparatuses and decrease damage risk (mechanical stress). Therefore, the simulations of two-dimensional axisymmetric models allow to describe the concentration distributions and the mass sources locations within porous electrodes. Momentum, heat, mass and charge balances were solved using COMSOL Multiphysics® commercial software. The general Butler–Volmer equation described the charge transfer at the triple phase boundary (TPB) of the anode/electrolyte interface. Only hydrogen oxidation was investigated due to the critical effect of water production (the cathodic reaction occurring in the blind part of the system). Simulations allowed calculation of distributions of velocity fields, thermal gradients and partial pressure distributions ($CH_4$, $H_2$, CO, $CO_2$, $H_2O$, $N_2$) within the anode. On the anode, the conversion process included methane reforming (dry and steam) associated with water–gas shift reactions. Computing results will be discussed assuming steam and dry reforming; in order to avoid lack of water vapor at the anode, humidification of inlet gas was achieved (1%).

## 2. Cell Design and Operating Conditions of Simulations

Tubular geometry was designed as and consisted of a cascade association of three conversion blocks. These blocks exhibited assembly of porous electrodes and porous electrolytes. The membrane electrode assemblies were spaced 1.5 mm apart; the second assembly featured a 22 mm diameter hole drilled in the center of the ceramic disc. The design of our concept is shown in Figure 1; the gas inlet is at the top of the cell and exhaust gas is in the bottom of cell. Conversion of methane into hydrogen occurred in a catalytic layer through polarization of the catalyst layer through the porous solid oxide electrolyte (Table S1). Therefore, the block included:

- A catalytic layer of cermet CGO–Ni cermet (CGO: cerium-gadolinium oxide) with doped Ir, noted as Ir–cermet;
- A porous membrane of YSZ (YSZ: yttria-stabilized zirconia).

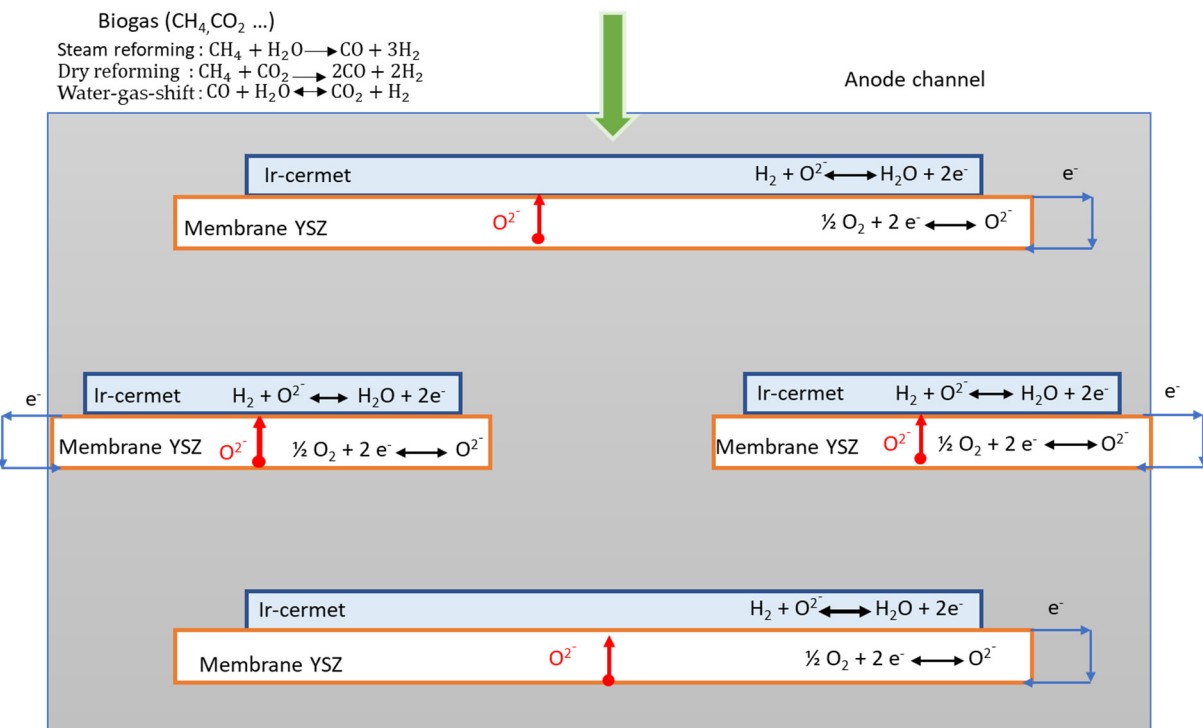

**Figure 1.** Concept of biogas converter using solid oxide cell.

Geometric parameters of the various elements are listed in Table 1 (exact geometry is available in Supplementary Materials Figure S1).

**Table 1.** Geometric parameters.

|  | Membrane | Catalyst Layer | Tube Size | Channel Height |
|---|---|---|---|---|
| Radius (mm) | 19/2 | 7 | 11 |  |
| Height (mm) | 1 | 0.1 | 26 | 1.5 |

In order to reduce and simplify computing time, the modeled geometry was axisymmetric. It was possible to consider only half of the studied system thanks to symmetry axes (Figure 1) [15]. Regardless of the size of the elements of this module, the porous cathode was a very thin component. Therefore, adaptation of the meshing was required in order to increase numerical accuracy and decrease machine–time costs (see in the Supplementary Materials the computations with operating parameters Tables S2–S7, the report of meshing sensitivity is available on Tables S8 and S9 and Figures S2–S5). Figure 2 shows mesh fitting of the studied geometry, and Table 2 gathers the numbers of nodes, the average surface and the statistic elements.

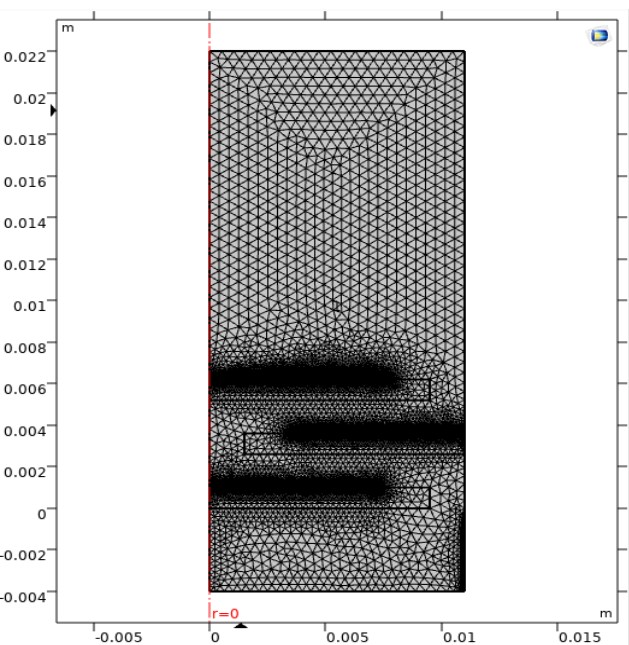

**Figure 2.** Triangular meshing using ComsolMultiphysics5.4.

**Table 2.** Nodes and statistics of meshed domain.

| Number of Elements | Minimum Element Quality | Average Element Quality | Element Area Ratio | Mesh Area (m$^2$) |
|---|---|---|---|---|
| 28,560 | 0.0896 | 0.8989 | $5.051 \times 10^{-5}$ | $2.86 \times 10^{-4}$ |

## 3. Model and Assumptions

In this study, the cells operated as half-cells for conversion of biogas and the corresponding balance equations of momentum, energy, mass and charge in a steady state were simultaneous solved. The purpose of this stationary model was to link these distributions to biogas convection behavior within the cells as well as possible. All equations and assumptions were based on state-of-the-art [12]. The kinetics of reactions used in this model included reversibility of chemical reactions taking place within the cell. The kinetics of the carbon deposition reaction were not taken into account thanks to Ir catalyst prevention [16]. Various reactions occur within a fuel cell; however, only five reactions have relatively large kinetic influences on the system. Therefore, each reaction is given in Table 3. Set parameters and all equations are available in the Supplementary Materials; only the main physics are detailed below.

**Table 3.** Reactions considered in cell.

| Reactions | Equations | Kinetics | Localizations |
|---|---|---|---|
| Steam Reforming | $CH_4 + H_2O \rightarrow CO + 3H_2$ | rvr (mol·m$^{-3}$·s$^{-1}$) | Ni-Ir Surface |
| Dry Reforming | $CH_4 + CO_2 \rightarrow 2CO + 2H_2$ | rdr (mol·m$^{-3}$·s$^{-1}$) | Ni-Ir Surface |
| Water–Gas shift | $CO + H_2O \rightarrow CO_2 + H_2$ | rgs (mol·m$^{-3}$·s$^{-1}$) | In Gas Phase |
| Hydrogen Oxidation | $H_2 + O^{2-} \rightarrow H_2O + 2e^-$ | $i_a$ (A·m$^{-3}$) | CGO-Cermet/YSZ Interface |

Cathodic electrochemical reaction was not considered, as it occurs outside of the biogas conversion cell.

### 3.1. Momentum Balance

Inlet feeding of cells takes place at the top of the system. To attain velocity distribution, Navier–Stokes momentum equations, expressed as follows, were solved:

$$\rho \frac{\partial u}{\partial t} - \nabla \cdot [\mu (\nabla u + (\nabla u)^T)] + \rho (u \cdot \nabla) u + \nabla p = F \tag{1}$$

$$\nabla \cdot u = 0 \tag{2}$$

This study was carried out in the steady-state condition, and volume force as well as inertial forces to viscous forces were nil ($(u \cdot \nabla) = 0$), so Equation (1) became

$$- \nabla \cdot [\mu (\nabla u + (\nabla u)^T)] + \nabla p = 0 \tag{3}$$

Several boundary conditions (BCs) were required to compute the multi-physic approach. The following BCs were considered:

Gas feeding (Figure 1) exhibits a very low speed: the flow will always be considered laminar, the inlet speed is defined as Hagen–Poiseuille velocity distribution.

Exhaust is defined as operating pressure: $p = p_{rf}$.

No-slip wall conditions are defined in other borders.

### 3.2. Mass Balance

The mass transfer phenomena involved convection and diffusion processes along with the chemical reaction. The equation describing the mass balance in gas phase is expressed as follows:

$$\nabla \cdot J_i + \rho (u \cdot \nabla) \, w_i = R_i \tag{4}$$

The velocity field $u$ (m·s$^{-1}$) was given by the Navier–Stokes equation (Equation (1)). In this numerical model, source terms were the reaction rates of each gaseous species, as summarized in Table 4.

**Table 4.** Expression of reaction rate of molar species present in fluid.

| Molar Reaction Speeds | |
|---|---|
| **Anode Side (kg·m$^{-3}$·s$^{-1}$) \*** | |
| $R_{CH4}$ | $(-rvr - rdr) \, MCH_4$ |
| $R_{CO2}$ | $(rgs - rdr) \, MCO_2$ |
| $R_{CO}$ | $(rvr - rgs + 2rdr) \, MCO$ |
| $R_{H2O}$ | $(-rvr - rgs + \frac{i_a}{2F}) \, MH_2O$ |
| $R_{H2}$ | $(3rvr + rgs + 2rdr - \frac{i_a}{2F}) \, MH_2$ |

\* Details of kinetic equations are available in Supplementary Materials Equations (S1)–(S6).

$J_i$ (kg·m$^{-2}$s$^{-1}$) is the diffusion flux of species $i$. Therefore, the gas mi5tures' diffusion flow (Equation (5)) was obtained from Stefan–Maxwell equations that took into account influence of diffusivity of each component (matrix-calculation of binary diffusivity) such as:

$$J_i = \rho w_i \sum_k D_{ik} d_k \tag{5}$$

The driving force for diffusion of the species $k$, $d_k$ (m$^{-1}$), was calculated according to proportion ($k$) of the species in the mixture according to the following expression:

$$d_k = \nabla x_k + \frac{1}{p_{r\acute{e}f}} [(x_k - w_k) \nabla p_{r\acute{e}f}] \tag{6}$$

Boundary conditions linked to mass balances were given as follows:
The inlet of the cell is at the top and the outlet at the bottom.
The porous ceramic membranes and their walls are permeable to fluids.

The electrochemical reaction is localized at the anode/membrane interface (molar flow density: internal boundary $\frac{i_a}{2F}$).

### 3.3. Heat Balance

As expected, temperature depended on many processes of the cell, such as kinetics of chemical and electrochemical reactions, the Joule effect in the porous ceramic and usual heat transfer phenomena. Definition of sources and heat transport is critical in order to predict chemical conversion behavior. A study by Daun et al. [17] has shown that exchanges of heat radiation are negligible compared to conduction and despite high temperatures. In this work, only convection and conduction phenomena were studied. Therefore, to achieve the temperature field within the cell, it was necessary to solve the heat equation, which is written as follows:

$$\rho C_P u \nabla T - \nabla \cdot (\lambda_{eff} \nabla T) = Q_c \tag{7}$$

Therefore, the source of total thermal energy of the system took the following form:

$$Q_c = E_{vr} + E_{dr} + E_{gs} + E_{ohm} \tag{8}$$

Expressions of different heat sources are listed in Table 5 (details of $E_{ohm}$.computations are available in Supplementary Materials Equations (S6)–(S12)). As far as the direction of reforming reactions, steam reforming and dry reforming are endothermic, and the water–gas shift reaction is exothermic. Accordingly, thermal sources have different origins. In order to define the model, knowing where to locate each source is important. The location of sources in chemical reactions is simple, since heat dissipation is found where those reactions take place: the surface of the catalyst layer for steam reforming and dry reforming and in whole device for the water–gas shift reaction. Electrochemical heat sources appear in many different forms, and sometimes, it is difficult to distinguish them. The main portion of heat is related to strength of materials in response to displacement of electronic and ionic species. This Ohmic loss, more commonly called the Joule effect, is located within the electrolyte. Finally, reaction enthalpies at 1123 K were calculated from the following expression:

$$\Delta_r H_T^{\circ} = \Delta_r H_{298}^{\circ} + \int_{298}^{T} \sum v_i C_{p_i} dT \tag{9}$$

and standard enthalpy of the reactions (T = 273 K and p = 1 bar) is given by

$$\Delta_r H_{298}^{\circ} = \sum v_i \Delta_f H_{i,T}^{\circ} \tag{10}$$

**Table 5.** Expressions of heat sources.

| Heat Sources | Expression (W/m³) | Location |
|---|---|---|
| Chemical Energies | | |
| Steam Reforming, $E_{vr}$ | $R_{vr}\Delta_r H_{vr}$ | Ir–CGO Cermet Volume |
| Dry Reforming, $E_{dr}$ | $R_{dr}\Delta_r H_{dr}$ | Ir–CGO Cermet Volume |
| Water-Gas Shift, $E_{gs}$ | $R_{gs}\Delta_r H_{gs}$ | In Gas Phase |
| Electrochemical Energies: in this work, the main part of the electrochemical irreversibility process was due to the Joule effect. | | |
| Joule Effect *, $E_{ohm}$ | $i\nabla V_{ion}$ | In YSZ Porous Material |

* Ionic potential distribution (charge balance) was solved using the Laplace equation (see in Supplementary Materials Equations (S6)–(S12)).

The thermo data and all parameter values of viscosity, diffusion coefficient, thermal conductivity and their dependence on temperature are available in the Supplementary Material

(Tables S1–S7 and Equations (S13)–(S32)) and based on literature [18–25]. Therefore, the heat flux according to the Fourier equation is given by:

$$\Phi = -\lambda \nabla T \qquad (11)$$

Basic boundary conditions were considered, such as:
The cell is simulated for a cross-flow supply, with inlet at the top and outlet at the bottom.
The cell walls are insulated.

## 4. Results and Discussion

The composition of feed materials respective to carbohydrate, fat and protein content impacted biogas yield and the methane amount in the biogas. This was verified with the theoretical biochemical methane potential calculated from the elemental chemical formulas [26]. When substrate or wastes with unknown organic compounds are digested, the ratio between chemical oxygen demand (COD) and total organic carbon (TOC) allows prediction of $CH_4$ content according to the following equation: $CH_4$ in % = 18.75 COD/TOC [27]. Therefore, various methanization routes have been considered in order to define inlet compositions of simulations. As quantity and quality of biogas production depend on different process parameters, including pH, temperature, substrate, loading rate, hydraulic retention time (HRT), C/N ratio and mixing, low methane content in A and B could correspond to specific situations where biogas production has been affected. For example, this has been observed in a dairy wastewater treatment plant including an unheated up-flow anaerobic sludge blanket (UASB) [28], or in co-digestion processes where protein and lipid percentages of particulate biodegradable COD are too high [29]. Therefore, due to the large variability of the $CH_4/CO_2$ ratio in Table 6, we selected some typical methane–carbon dioxide compositions. Steam content was fixed equal to 1%, and a neutral gas ($N_2$) brought each mixture to 100%. The results of syngas production as a function of biogas sources are gathered in Table 6, including rate of production ($H_2$ & CO) and rate of conversion ($CH_4$ & $CO_2$).

**Table 6.** Biogas conversion rate depending on feed composition (inert gas used to complete mixture).

| Mass Fraction of Mixture % CH₄:CO₂:H₂O | | Consumption Rate % | | Production Rate % | |
|:---:|:---:|:---:|:---:|:---:|:---:|
| | | $\Delta CH_4$ | $\Delta CO_2$ | $\Delta H_2$ | $\Delta CO$ |
| A | 28:70:1 | 99.99 | 63.64 | 99.37 | 99.99 |
| B | 34:62:1 | 98.9 | 75.85 | 99.48 | 99.99 |
| C | 40:50:1 | 90.99 | 85.22 | 99.5 | 99.99 |
| D | 49:49:1 | 82.85 | 87.24 | 99.51 | 99.99 |
| E | 50:40:1 | 79.31 | 94.2 | 99.48 | 99.99 |
| F | 60:39:1 | 72.87 | 96.1 | 99.5 | 99.99 |
| G | 70:28:1 | 54.66 | 99.99 | 99.37 | 99.99 |

In Table 6, the production rate of "syngas" is significantly high regardless of composition of the incoming biogas. However, regarding the conversion rate of biogas for an inlet composition containing predominantly methane (case G), the conversion rate of the case G was only half the value of that obtained with poor composition of inlet methane (case A). Consequently, the carbon dioxide's rate of consumption or conversion was almost 100% (case G). $CO_2$ utilization was promoted with any methane composition, but only 40% of methane amount (case C) was required to attain 85% of $CO_2$ conversion. Maximum conversion in syngas is achieved when the proportion of methane and carbon dioxide in the mixture is almost the same (as in cases C, D and E). For efficient scrutinization of conversion, partial-pressure profiles of hydrogen, carbon monoxide, methane and carbon dioxide

within the cell are exhibited in Figure 3, with various inlet gas compositions (Figure 3): A, C and G (Table 6).

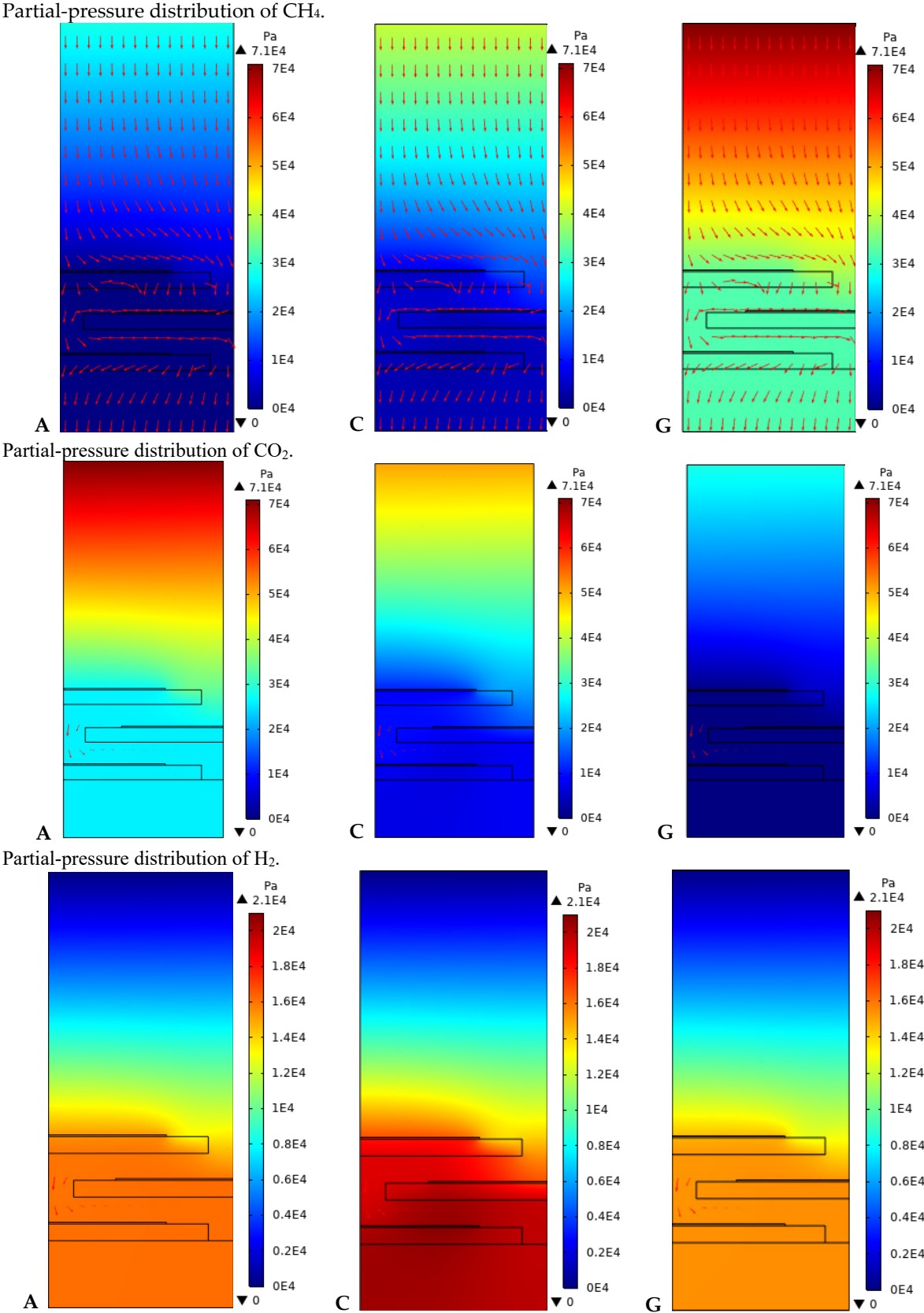

**Figure 3.** *Cont.*

Partial-pressure distribution of CO.

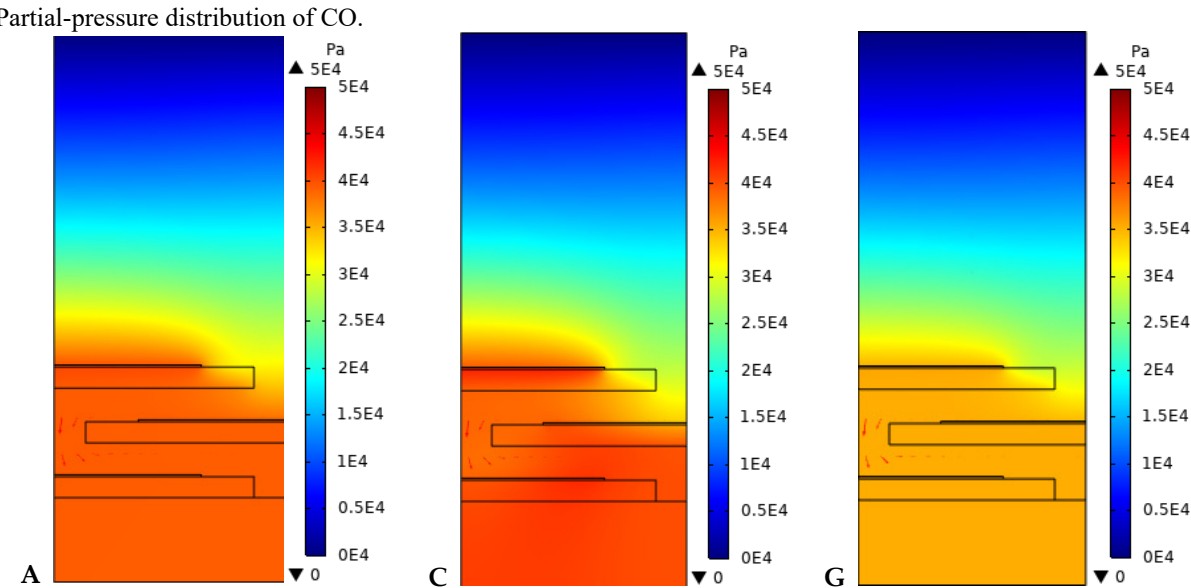

**Figure 3.** Partial-pressure distributions of $CH_4$, $CO_2$, $H_2$ and CO in cases A, C and G.

Gas distribution showed that an equal proportion of methane and carbon dioxide in the inlet mixture allowed higher conversion. Indeed, the partial pressures of hydrogen and carbon monoxide at the cell outlet were noticeably higher in the "C" conditions, where the average conversion rate of the biogas was equal to 88.1%. In the "C" inlet condition, greater production of hydrogen and high conversion of methane were observed. The composition with 40% CH4, 50% $CO_2$ and 1% steam seemed to be the optimum of syngas production from biogas via this electrochemical device.

In order to assess the effect of the reaction kinetics, only case "C" was acutely studied. The catalytic layers of the cell are numbered from top to bottom: CELL 1 to CELL 3 (Figures 4 and 5).

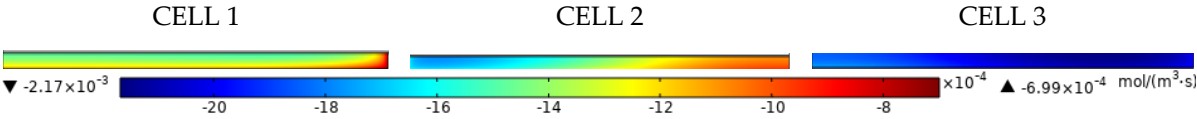

**Figure 4.** Reaction rate distribution of steam reforming along catalyst layers from top to bottom in mol/($m^3$·s).

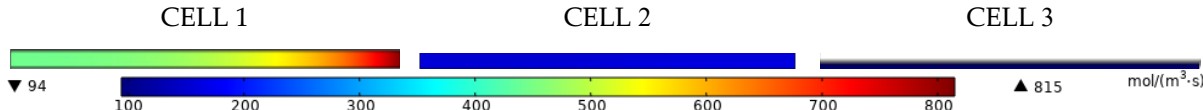

**Figure 5.** Reaction rate distribution of dry reforming along catalyst layers from top to bottom in mol/($m^3$·s).

Steam-reforming and dry-reforming reactions take place on the surface of catalytic grains along the catalytic layer. The steam-reforming kinetic was almost constant along the first catalytic layer (1), while on the second layer (2), the gradient was a more marked, and the third (3) had a constant, lower kinetic. The negative weak value of the vapor-reforming kinetic reflected predominance of dry reforming due to temperature distribution (Figure 6). In addition, variation of steam-reforming kinetics between each catalytic layer was due to methane pressure depletion; considerably fewer reagents ($H_2O$ & $CH_4$) were available to carry out the steam-reforming reaction. In these simulations, dry-reforming kinetics were large in the first cell and less for each successive cell (see additional results in Supplementary Materials, Figures S6–S12). The kinetics of the steam-reforming reaction

were negligible compared with those of the dry reforming process; this high speed thus allowed production of CO and $H_2$, and those results are in line with the thermodynamic description from D. Papurello et al. [12]. The hydrogen produced was consumed by the electrochemical reaction in order to produce water, which was consumed by the steam-reforming reaction. The kinetics of the gas-shift reaction were negative (Figure 6), implying that it would consume hydrogen and carbon dioxide in order to produce water vapor and carbon monoxide; this value led to nil in the system. However, due to the decrease of temperature (Figure 6) along the gas channel, the water–gas shift reaction was dumped in order to maximize the conversion of $CO_2$ into CO. Local production of water thanks to the electrochemical reaction and fast temperature depletion act positively towards syngas production. Therefore, specific attention was paid to the solid oxide cell temperature behavior in order to avoid excessive temperature gradients, which would not have been tolerated by the oxide materials [30]. In the present study, these gradients were oriented from the inlet to the outlet in order to avoid damages on each membrane electrode assembly. The two-dimensional simulations that were performed in this study could be updated with new phenomena. The governing equations shall to be modified in order to account for performance degradation due to trace amounts of fuel contaminants (phosphine) [31] or to thermal damage [30]. In addition, recent studies have highlighted that the computing methods are powerful tools: e.g., the simulations using Aspen Plus that were performed by R. S. Cavaignac et al. [32], which evaluated different process conditions of biogas purification, or the work of Papurello et al. [12], where the CFD computations in a SOFC tubular cell system (direct biogas feeding) were achieved.

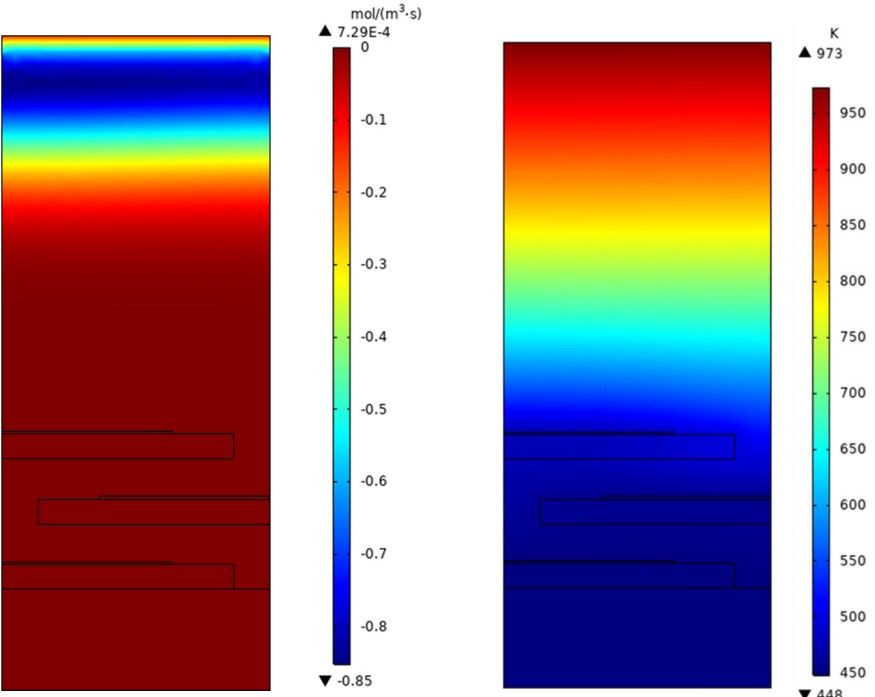

**Figure 6.** Distribution of reaction rate of water–gas shift in $mol/(m^3 \cdot s)$ and temperature (K) in case C.

## 5. Conclusions

The computer-aided engineering approach made it possible to achieve a virtual proto-type and describe expected performances. In this study, simulation results highlighted that production of syngas was possible through supplying of an SOFC cell made up of a porous membrane, with biogas mainly comprising methane and carbon dioxide. Therefore, dry re-forming of biogas appears to be an interesting way to avoid the cost of biogas purification [32].

Composition of the biogas mixture (depending on the production process) at the inlet of the device strongly influenced the conversion rates of $CO_2$ and $CH_4$. The production of syngas was optimal with a $CO_2/CH_4$ ratio = 1 at inlet. In addition, the configuration of the

cascade conversion cell in the direction of the supply of the cell resulted in larger active surface areas for the steam-reforming reaction. This was not limited by the low proportion of water in the mixture because the electrochemical reaction allowed it to be produced in sufficient quantities to carry out the steam-reforming reaction. Moreover, the temperature gradients in the same direction as the flow rate should limit thermal stress in oxide material and could void the reverse-gas-shift reaction.

Future works will focus on current collectors and a degradation model for application of fuel contaminants such as phosphine.

**Supplementary Materials:** The following supporting information can be downloaded at: https://www.mdpi.com/article/10.3390/hydrogen3040031/s1, Figure S1. Boundary layers mesh. Figure S2. Free quad mesh. Figure S3 Hydrogen partial pression obtained using boundary layers mesh (left) using Free Quad mesh (right). Figure S4. Methane partial pression obtained using boundary layers mesh (left) Methane partial pression obtained using Free Quad mesh (right). Figure S5. Geommetry of simulated cells. Figure S6. Velocity, partial pressure of steam for C case. Figure S7. Velocity, partial pressure of steam for C case. Figure S8. Reaction rate distribution of Steam and dry reforming along the catalyst layers from top to bottom in mol/(m$^3$.s) (A). Figure S9. Velocity, partial pressure of steam for F case. Figure S10. Reaction rate distribution of Steam and dry reforming along the catalyst layers from top to bottom in mol/(m$^3$.s) (F). Figure S11. Velocity, partial pressure of steam for G case. Figure S12. Reaction rate distribution of Steam and dry reforming along the catalyst layers from top to bottom in mol/(m$^3$.s) (G). Table S1. Intrinsic parameters of materials microstructure. Table S2. Values of kinetic parameters of reactions. Table S3. Viscosity parameters. Table S4. Diffusion parameters. Table S5. Stefan-Maxwell matrix. Table S6. Expression of thermal conductivity of species according to the temperature [8]. Table S7. Thermal capacity of endangered gas phase according to the temperature. Table S8. Nodes and statistics of meshed domain: mesh boundaries layers (triangles). Table S9. Nodes and statistics of meshed domain: mesh Free quad.

**Author Contributions:** Conceptualization, J.D.; methodology, J.D. & K.P.K.; software, K.P.K. & J.D.; validation, K.P.K. & J.D.; formal analysis, J.D., G.B., G.M.; investigation, G.M. & J.D.; resources, J.D.; data curation, G.M. & J.D.; writing—original draft preparation, K.P.K., G.M. & J.D.; writing—review and editing, F.D., G.B., G.M. & J.D.; visualization, K.P.K.; supervision, J.D. & G.M.; project administration, J.D. All authors have read and agreed to the published version of the manuscript.

**Funding:** This research received no external funding.

**Institutional Review Board Statement:** Not applicable.

**Informed Consent Statement:** Not applicable.

**Data Availability Statement:** The computing model file has been shared as Supplementary Materials.

**Conflicts of Interest:** The authors declare no conflict of interest.

## Nomenclature

| | |
|---|---|
| $i$ | Total current density of cell, $A \cdot m^{-2}$ |
| $i_a$ | Anodic current transfer, $A \cdot m^{-3}$ |
| $p$ | Pressure, Pa |
| $p_{rf}$ | Atmospheric pressure, Pa |
| $p_i$ | Partial pressure species $i$, Pa |
| $rdr$ | Kinetic of dry reforming, $mol \cdot m^{-3} \cdot s^{-1}$ |
| $rgs$ | Kinetic of water–gas shift, $mol \cdot m^{-3} \cdot s^{-1}$ |
| $rvr$ | Kinetic of steam reforming, $mol \cdot m^{-3} \cdot s^{-1}$ |
| $u$ | Rate, m/s |
| $w_i$ | Mass fraction of species $i$ |
| $x_i$ | Molar fraction of species $i$ |
| $C_{p_i}$ | Molar heat capacity of species at constant pressure, $J \cdot mol^{-1} \cdot K^{-1}$ |

| | |
|---|---|
| $D_{ik}$ | Binary coefficient diffusion, $m^{-2} \cdot s^{-1}$ |
| $E_{dr}$ | Heat generated from dry reforming, $W/m^3$ |
| $E_{gs}$ | Heat generated from water–gas shift, $W/m^3$ |
| $E_{vr}$ | Heat generated from steam reforming, $W/m^3$ |
| $E_{rev}$ | Heat generated from electrochemical reaction, $W/m^3$ |
| $E_{ohm}$ | Heat generated from Joule effect of YSZ, $W/m^3$ |
| $J_i$ | Molar flux density of species $i$, $kg \cdot m^{-2} s^{-1}$ |
| $M_i$ | Molecular weight of species $i$, $g/mol$ |
| $R_i$ | Mass sources of species $i$, $kg \cdot m^{-3} s^{-1}$ |
| $Q_c$ | Heat sources, $W/m^3$ |
| $\lambda$ | Thermal conductivity, $mW \cdot m^{-1} \cdot K^{-1}$ |
| $\mu$ | Dynamic viscosity, $Pa \cdot s$ |
| $\rho$ | Density, $kg \cdot m^{-3}$ |
| $\phi$ | Heat flow, $W \cdot m^{-2}$ |
| $\Delta_r H_T^{\circ}$ | Reaction enthalpy at temperature $T$, $J/mol$ |

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
