# Peer review of "Simulation of Biogas Conversion Using Porous Solid Oxide Electrochemical Cells: Virtual Prototyping"

_hydrogen, doi:10.3390/hydrogen3040031_

Round 1

Reviewer 1 Report

The authors used modeling to investigate the distributions of partial pressures for all gas species within the anode (CH4, H2, CO, CO2, H2O, N2), velocity field and temperature in a cascade conversion cell for syngas production from biogas. The steam and dry reforming associated with water gas shift reaction are considered during the conversion process. The content fits for the Hydrogen journal, while there are several critical issues with this manuscript. Please see comments below.

1.       Some grammar issues in the manuscript. For example, the sentence from line 25-28, line 29-31, etc. In addition, some typos in the manuscript, for example the first letter should be capital for the beginning of a sentence on line 48, 111, etc. Please double check throughout the whole manuscript to avoid such issues.

2.       Introduction section, the authors haven’t pointed out the need of this study. Please revise the introduction section to highlight the importance of this study and also the novelty of this work. In addition, the more related works should be cited.

3.       The full names of CGO and YSZ are missing.

4.       Figure 1 is very confusing. It would be better to show the parameters in Table 1 and the spaced size 1.5 mm directly in Figure 1. In addition, the authors should provide a detailed description of the structure of the cell. BTW, what it the external connection of this special three-electrode half-cell, which should also be included in Figure 1?

5.       For the mesh quality, have you checked the impact on modeling results?

6.       Have you validated the model? Without the model validation, its results are doubtful. Please add this part.

7.       The color scale bar in Figure 3 and 6 is misarranged.

8.       Results seem too less. The authors only presented the distribution of partial pressures, and reaction rate. The deep analysis is missed. What is the main findings?

Author Response

Dear Reviewer a point-by-point answers to your questions/comments and the related modifications performed in the manuscript are enclosed below. We have addressed and incorporated all your comments and believe that they have helped to improve the quality of this paper.

Reviewer 2 Report

GENERAL:

·       This study deals with the direct production of syngas with biogas using the configuration of the cascade conversion cell in the supply feed direction of the system. Commercial software COMSOL Multiphysics®  was used in the simulation study. In my view, some essential concerns should be addressed for improvement.

SPECIFIC COMMENTS:

ABSTRACT:

The authors only state that Commercial software COMSOL Multiphysics®  has been used to explain the research methodology. There is no further explanation of the research scenario in the abstract. What about the numerical simulation that was realized? What about the quality of the mesh? Besides, it would be better to state the problem statement with a brief background in the early abstract.

INTRODUCTION:

·       The authors have presented the introduction. The authors should emphasize and explain the current study's novelty, which differs remarkably from previous research. Why is it important?

MATERIAL AND METHODS:

·       The simulation applied triangular meshing using Comsol Multiphysics 5.4. My question is, “what about the quality of the mesh”? How to measure whether the simulation data is adequate using the mesh created? There was no data validation step here. The authors need to explain the validation process. The authors can complete the comparison data from other researchers for validation (if available).  

·       In my view, the use of rectangular structured mesh seems to be able to maintain the consistency of the mesh more than the unstructured mesh form.

·       In addition, it is helpful to complete the description of collecting data, data processing scenarios, and how to interpret the data collection.

RESULTS & DISCUSSION:

·       Most of the research results are displayed using contour gradations, and it would be more interesting to provide results using graphs or other expressions for variation. Thus, the author can show the results quantitatively compared to other research for in-depth discussion.  

·       It would be more meaningful to explain why these findings are essential in the field and their conclusions and principal implications. 

Author Response

(The authors gave the same response as above.)

Round 2

Reviewer 1 Report

This is a revised version and the authors have addressed all the comments from previous reviewers. The manuscript is in good shape now.

Author Response

We thank the reviewer for his comments and questions which greatly contributed to improving the manuscript.

Reviewer 2 Report

The author has made several revisions according to the comments, although not completely perfect. In the future, there needs to be more advanced writing by considering the structured mesh to control the resolution quality of the mesh in the area near the wall. Thus the results of the prediction data can be more optimal.

Author Response

We agree with the reviewer the structured mesh to control the resolution quality of the mesh close to the boundaries is substantially more efficient. In our future work we will make sure to seek more accuracy to our  next simulations.

We thank the reviewer for his comments and questions which greatly contributed to improving the manuscript.